# DiffusionRet: Diffusion-Enhanced Generative Retriever using Constrained Decoding

**Shanbao Qiao** and **Xuebing Liu** and **Seung-Hoon Na**
Center for Advanced Image and Information Technology,
Department of Computer Science and Artificial Intelligence,
Jeonbuk National University
{joe,liuxuebing,nash}@jbnu.ac.kr

## Abstract

Generative retrieval, which maps from a query to its relevant document identifiers (docids), has recently emerged as a new information retrieval (IR) paradigm, however, having suffered from 1) the *lack of the intermediate reasoning step*, caused by the manner of merely using a query to perform the docids generation or hierarchical classification, and 2) the *pretrain-finetune discrepancy*, coming from the use of the artificial symbols of docids. To address these limitations, we propose a novel approach of using document generation from a query as an intermediate step before retrieval, thus presenting diffusion-enhanced generative retrieval (**DiffusionRet**), which consists of two processing steps: 1) *diffusion-based document generation*, which employs the sequence-to-sequence diffusion model to produce a pseudo-document sample from a query and is expected to be semantically close to a relevant document. 2) *N-gram-based generative retrieval*, which uses another sequence-to-sequence model to generate n-grams appearing in the collection index for linking a generated sample to an original document. Experimental results on the MS MARCO and Natural Questions datasets show that the proposed DiffusionRet significantly outperforms all existing generative retrieval methods, leading to state-of-the-art performance, even with much a smaller number of parameters. [1]

## 1 Introduction

*Generative retrieval* methods have emerged as powerful IR paradigms (Tay et al., 2022; Zhou et al., 2022; Wang et al., 2022; Chen et al., 2022a; Ren et al., 2023; Bevilacqua et al., 2022; Tang et al., 2023a), by employing sequence-to-sequence models to map a query to relevant docids, showing promising results compared with *dense retrieval* based on dual encoders (Karpukhin et al., 2020; Ni

et al., 2022). In generative retrieval, because all the required information and knowledge for indexing and retrieval are maintained in parametric memory, retrieval is conducted efficiently at the inference stage by a decoder, without requiring other inverted index structures or tools for the nearest neighbor search, such as FAISS (Johnson et al., 2019).

However, the classical methods of existing generative retrieval mostly produce only docids at inference time, much shorter than normal sentences, while no other content is generated (Tay et al., 2022; Zhuang et al., 2022; Wang et al., 2022). *Lacking an intermediate reasoning step*, it is unclear what detailed reasoning steps proceed during decoding-based hierarchical classification [2] or docids generation, which is likely far from the improved approach based on the chain-of-thought (Wei et al., 2022; Trivedi et al., 2023). Additionally, generative retrieval extensively uses artificial symbols of docids for target sequences. As noted in (Ren et al., 2023), however, these docid-like symbols are only seen during the finetuning but are unseen at the pretraining stage, thus causing the *pretrain-finetune discrepancy*, largely limiting the effect of transfer learning, which is arguably important when exploiting pretrained models.

To address these limitations, we establish the document generation stage as a key intermediate reasoning step, that directly generates the "actual content" of a document relevant to a given query. Once a document is generated, we use it as a new document-level query, generate N-grams restricted to those occurring in the collection, and use them to identify its original source document. To gen-

---

[1] Our code and models are available at https://github.com/Miracle0814/DiffusionRet.

[2] Here, we primarily focus on the classical type of generative retrieval (Tay et al., 2022; Zhuang et al., 2022; Wang et al., 2022) where the hierarchical clustering process is deployed to assign semantically structured docids, and interprets its generation of a sequence of docids as the hierarchical classification, strongly based on (Risch et al., 2020), which views the hierarchical document classification as a sequence-to-sequence task.

erate such a pseudo query-relevant document, the diffusion model emerges as a noticeably promising alternative to the parameter-intensive large language models (LLM) (Li et al., 2022b), surpassing some GPT variants even with a much smaller parameter scale, as in (Gong et al., 2023; Yuan et al., 2022; Gao et al., 2022b; Li et al., 2023a; Tang et al., 2023b; Bi et al., 2023). Pursuing the use of the *reasoning chain of document-ngrams* as an intermediate step, inspired by the aforementioned notable parameter efficiency of diffusion models, we propose a two-stage diffusion-enhanced generative retrieval (**DiffusionRet**) model, which consists of two processing steps, as follows:

- **Diffusion-based document generation**, which extensively applies the sequence-to-sequence diffusion model (Gong et al., 2023; Yuan et al., 2022) to generate pseudo-documents for an input query, similar to text-based *semantic inpainting*, where full contents of a document are infilled given a query as a context. Although a pseudo-document is not the same as the original one, it is expected that it will look like real documents in length, format, and content, and contain semantically enriched content of an original query. By providing a useful intermediate representation with this enriched information, a generated pseudo-document allows us to perform in-depth semantic matching at the document-to-document level in subsequent retrieval. Furthermore, like (Ren et al., 2023), the *data distribution mismatch* between the training and inference, an issue faced by the classical generative retrieval, is somehow effectively handled by putting the 'intermediate reasoning step", as the subsequent retrieval phrase takes not queries but extended pseudo-documents produced by the diffusion model, whose lengths and formats exhibits the regularity, aligning closely with those observed during indexing phrase.

- **N-gram-based generative retrieval**, which takes a pseudo-document resulting from the diffusion model and generates n-grams based on (Bevilacqua et al., 2022), another sequence-to-sequence model where decoding is restricted to the set of FM-indexed n-grams in the collection. Given the experimental evidence of the effect of n-gram length in

(Bevilacqua et al., 2022) that shorter n-grams are less informative, we empirically consider long n-grams with a length of 30, more likely to be unique to a specific document [3]. Using n-gram identifiers, our generative retrieval model does not suffer from the pretrain-finetune discrepancy because special tokens such as docids are not necessary.

To the best of our knowledge, this is the first study where the diffusion model is deployed to generate a realistic document sample for the IR task.

Our contributions are summarized as follows: 1) we propose DiffusionRet, a two-stage generative retrieval model based on the reasoning chain of document-ngram, without requiring special tokens such as docids, 2) we apply a diffusion model for document generation as a textual version of the semantic inpainting, 3) The proposed DiffusionRet shows a state-of-the-art performance on the MS MARCO and Natural Questions (NQ) datasets.

## 2 Related Works

### 2.1 Diffusion Models for Text Generation

Diffusion models (Sohl-Dickstein et al., 2015; Song and Ermon, 2019; Ho et al., 2020; Yang et al., 2022) have emerged as new and prominent approaches exhibiting remarkable performances such as the generation of high-quality images, particularly in computer vision tasks (Saharia et al., 2022; Li et al., 2022a; Lugmayr et al., 2022). Recently, diffusion models have been extensively studied for text generation tasks (Li et al., 2022b; Gong et al., 2023; He et al., 2022; Yuan et al., 2022; Ye et al., 2023). DiffusionLM (Li et al., 2022b) presents one of the earlier diffusion models for controlled text generation, but its controlling settings are limited to simplified conditions, such as parts-of-speech and syntax trees. DiffuSeq (Gong et al., 2023) extends the diffusion model to sequence-to-sequence tasks, enabling fine-grained controlled generation using a text source as the control condition. More recently, SeqDiffuSeq (Yuan et al., 2022) extended DiffusionLM to a sequence-to-sequence setting by equipping it with a self-conditioning technique (Chen

---

[3]We do not claim that using long lengths of n-grams leads to the improvement in the retrieval performance. Instead, the n-gram-based generative retrieval does not need the requirement of the "extensive data augmentation" like query expansion, which is present in SEAL (Bevilacqua et al., 2022), possibly due to our two-stage setting that handles the data distribution mismatch between training and inference.

et al., 2022b; Strudel et al., 2022) and by proposing a novel adaptive noisy schedule. In this study, we extensively utilize SeqDiffuSeq to generate document samples for the IR problem, because it is a more advanced variant, whereas other advanced diffusion models could be deployed in our proposed method.

## 2.2 Generative Retrieval

Generative retrieval (Metzler et al., 2021; Tay et al., 2022; Wang et al., 2022; Sun et al., 2023; Zhuang et al., 2022; Tang et al., 2023a; Zhou et al., 2022; Bevilacqua et al., 2022; Chen et al., 2022a, 2023; Ren et al., 2023), also called the Differentiable Search Index (DSI) (Tay et al., 2022), has emerged as a new IR framework, integrating both the indexing and retrieval in a single text-to-text transformer without requiring an external index. Starting with the pioneering work of DSI (Tay et al., 2022) which trained T5 models to directly generate docids from a query, some studies adopted separate query generation modules to store mapping knowledge in the model parameters (Zhuang et al., 2022; Wang et al., 2022).

Instead of using a sequence of docids for each document, DynamicRetriever (Zhou et al., 2022) proposed a method that leverages the document embeddings resulting from the pretrained dual encoder to initialize the output embeddings of docids in the decoder and then finetunes the encoder-decoder to capture the query-docid relations.

Numerous studies have used semantically interpretable docids, without relying on artificial tokens for docids (Tang et al., 2023a; Bevilacqua et al., 2022; Chen et al., 2022a, 2023; Ren et al., 2023). SE-DSI (Tang et al., 2023a) proposed semantically more meaningful docids based on query generation and adopted rehearsal strategies to better associate a document with its docids. There are also some recent works based on pre-trained language models. SEAL (Bevilacqua et al., 2022) generates n-grams based on the FM-index (Ferragina and Manzini, 2000) to constrain the decoding of the autoregressive model to restrict the search space of the n-grams. CorpusBrain (Chen et al., 2022a) uses the Wikipedia article titles as docids to perform multiple pre-training tasks to encode knowledge in the corpus for handling a variety of downstream knowledge-intensive tasks. UGR (Chen et al., 2023) exploits n-gram-based identifiers to unify different retrieval tasks into a single genera-

tive form and employs a prompt learning strategy to enable better generalization across different tasks. MINDER (Li et al., 2023b) introduced novel synthetic identifiers generated from a passage's content and proposed multi-view identifiers based on three different types – titles, sub-strings, and synthetic identifiers (i.e., pseudo-queries) – to facilitate the holistic views of passages.

For document-level generation, query2doc (Wang et al., 2023) proposes a query expansion method by few-shot prompting LLMs to generate pseudo-documents. However, the performance of query2doc is significantly influenced by the model size, as they mentioned that the model shows the best when combined with the largest LLMs (i.e., 175B) while small models yield only slight improvements. Unlike query2doc with this dependence on LLMs, our diffusion-based generation enables to generate high quality pseudo-document with the considerably fewer model parameters.

The work most similar to ours is TOME (Ren et al., 2023), which also proposes a two-stage approach by first generating pseudo-documents. However, their second-stage retrieval deployed Web URLs as docids, which are semantically less meaningful than our use of n-grams. Moreover, we use diffusion models for document generation, which significantly reduce the parameter size (i.e., 103M parameters), compared to their case, which uses either T5-3B or T5-large. The experimental results show that DiffusionRet leads to noticeable improvements over TOME in the MS MARCO dataset, as discussed in Section 4.

## 3 Method

Figure 1 presents the overall architecture of DiffusionRet used to form a document-ngram reasoning chain, consisting of 1) diffusion-based document generation and 2) ngram-based generative retrieval. Formally, suppose that a query is given as $\mathbf{x}$, $\mathcal{C} = \{d_1, \cdots, d_N\}$ is a collection of documents where $d_i$ is $i$-th document, $\mathcal{V}$ is a vocabulary set of tokens, $n$ is the maximum length of the sequence, and $k$ is the fixed length of n-grams to be generated. DiffusionRet sequentially performs $\mathsf{Diffu}(\cdot)$, the diffusion-based document generation, and the n-gram generation, $\mathsf{Gen}_{ngram}(\cdot)$ to compute the

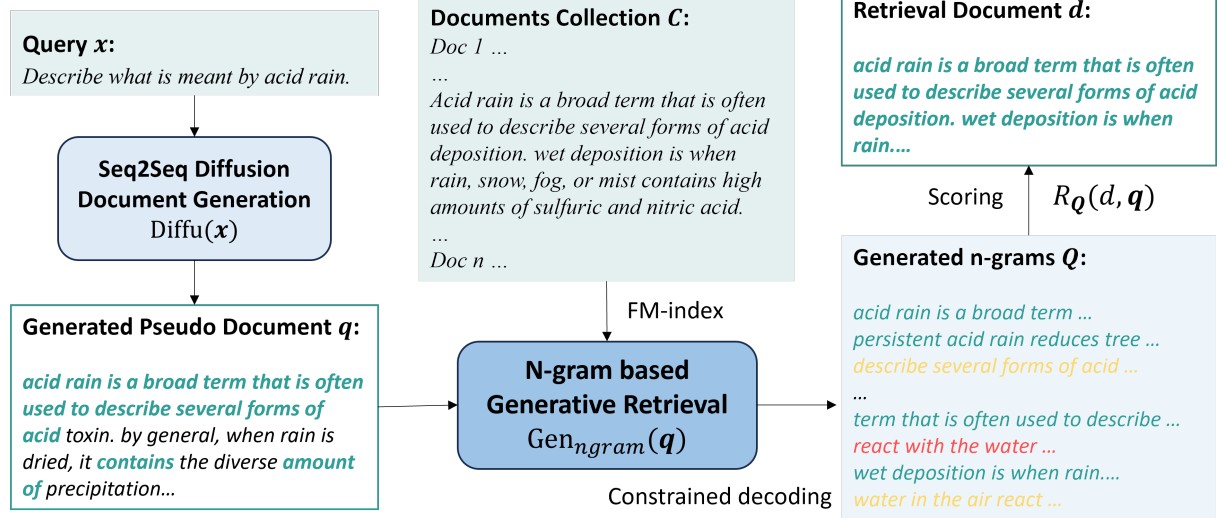

Figure 1: An overall architecture of the proposed DiffusionRet, formulated in Eq. (1): i) Given a query $\mathbf{x}$, the diffusion-based document generation, denoted as $\text{Diffu}(\mathbf{x})$, repeatedly applies the reverse denoising function of Eq. (3) and finally the rounding function to generate a pseudo-document $\mathbf{q}$ (i.e., Eq. (4)). ii) The N-gram generative retrieval applies $\text{Gen}_{ngram}(\mathbf{q})$ to generate a set of n-grams $\mathcal{Q}$ from $\mathbf{q}$, under the restricted space of the FM-index of the collection and then computes the relevance score $\text{R}_{\mathcal{Q}}(d, \mathbf{q})$ as in Eq (6).

relevance score of a document $d \in \mathcal{C}$ as follows:

$$
\begin{aligned}
\mathbf{q} &\sim \text{Diffu}(\mathbf{x}) \\
\mathcal{Q} &= \text{Gen}_{ngram}(\mathbf{q}) \\
Score(d, \mathbf{x}) &= \text{R}_{\mathcal{Q}}(d, \mathbf{q})
\end{aligned} \tag{1}
$$

where $\text{Diffu}(\cdot)$ is the reverse process of the sequence-to-sequence diffusion model of (Yuan et al., 2022), as described in Section 3.1, $\text{Gen}_{ngram}(\cdot)$ and $\text{R}_{\mathcal{Q}}(\cdot)$ are the constrained decoding process and the scoring function of (Bevilacqua et al., 2022), respectively, as described in Section 3.2. The generated document and n-grams (i.e., $\mathbf{q}$ and $\mathcal{Q}$) are used as key representations for intermediate reasoning in DiffusionRet.

### 3.1 Diffusion-based Document Generation

For $\text{Diffu}(\mathbf{x})$, we deploy SeqDiffuSeq (Yuan et al., 2022), a variant of the sequence-to-sequence diffusion model, for controlled generation at a given $\mathbf{x}$. As illustrated in Figure 2, the diffusion process consists of 1) the forward diffusion process, which gradually adds random noise to fit into a standard Gaussian distribution at the final time step, and 2) the reverse denoising process, which gradually denoises random noise to generate a synthetic sample.

**Forward process**  Suppose a document in $\mathcal{C}$ to be diffused is provided as an array of one-hot vectors,

i.e., $\mathbf{d} \in \mathbb{R}^{n \times |\mathcal{V}|}$ [4]. Unlike continuous diffusion models, $\mathbf{d}$ is first converted to a list of continuous vectors on the embedding spaces using $\text{Emb}(\mathbf{d})$, as adopted by (Li et al., 2022b):

$$
q(\mathbf{y}_0|\mathbf{d}) = \mathcal{N}(\mathbf{y}_0; \text{Emd}(\mathbf{d}), \beta_0 \mathbf{I}) \tag{2}
$$

where $\beta_t$ is a hyperparameter referring to the amount of noise added at the time step $t$, $\text{Emb}(\mathbf{d})$ is a trainable embedding function defined as $\text{Emb}(\mathbf{d}) = [\text{Emb}(\mathbf{d}_1), \cdots, \text{Emb}(\mathbf{d}_n)]$, in which $\mathbf{d}_i$ refers to $i$-th token of $\mathbf{d}$. The diffusion process of the subsequent time steps is the same as the standard one (Li et al., 2022b); for each time step $t \in \{1, \cdots, T\}$, the diffusion injects the random noise using the distribution $q(\mathbf{y}_t|\mathbf{y}_{t-1}) = \mathcal{N}(\mathbf{y}_t; \sqrt{\alpha_t}\mathbf{y}_{t-1}, (1 - \alpha_t)\mathbf{I})$.

**Reverse process**  In the reverse process, the denoising model generates a sample $\mathbf{y}_{t-1}$ conditioned on both $\mathbf{y}_t$ and $\mathbf{x}$, using the denoising distribution $p_\theta$ as follows:

$$
\begin{aligned}
p_\theta(\mathbf{y}_{t-1}|\mathbf{y}_t, \mathbf{x}) &= \mathcal{N}(\mathbf{y}_{t-1}; \mu_\theta(\mathbf{y}_t, \mathbf{x}, t), \beta_t \mathbf{I}) \\
\mu_\theta(\mathbf{y}_t, \mathbf{x}, t) &= A_t \text{ED}_\theta^0(\mathbf{y}_t, \mathbf{x}, t) + B_t \mathbf{y}_t
\end{aligned} \tag{3}
$$

where $\text{ED}_\theta^0(\mathbf{y}_t, \mathbf{x}, t)$ is the *denoising* function based on the encoder-decoder architecture, that

---

[4] Here, $\mathbf{d}$ is a version of one-hot vectors for $d_i \in \mathcal{C}$

takes $\mathbf{x}$ in the encoder and denoises the noisy output embeddings $\mathbf{y}_t$ in the decoder, and $A_t$ and $B_t$ are functions of $\alpha_1^t$ and $\beta_t$, defined in Appendix B.

Importantly, when $t = 0$, the *rounding mechanism* (Li et al., 2022b) is applied on the generated hidden vector $\tilde{\mathbf{y}}_0$, which is mapped back to the nearest token in the embedding space by the *rounding function* $\mathsf{Round}(\tilde{\mathbf{y}}_0) = \text{argmax}_{\mathbf{d}} p_\theta(\mathbf{d}|\tilde{\mathbf{y}}_0)$ where $p_\theta(\mathbf{d}|\tilde{\mathbf{y}}_0)$ is the multiplications of the softmax distributions over tokens after projecting $\tilde{\mathbf{y}}_0$ using the embedding matrix, and $\mathbf{d}$ is a list of one-hot vectors.

In summary, $\mathsf{Diffu}(\mathbf{x})$ is the function that repeatedly applies the denoising steps and rounding mechanism at the final timestep, as follows:

$$
\begin{aligned}
\tilde{\mathbf{y}}_T &\sim \mathcal{N}(\mathbf{0}, \mathbf{I}) \\
\tilde{\mathbf{y}}_{t-1} &\sim p_\theta(\mathbf{y}_{t-1}|\tilde{\mathbf{y}}_t, \mathbf{x}) \\
\mathbf{q} &= \mathsf{Round}(\tilde{\mathbf{y}}_0)
\end{aligned}
\tag{4}
$$

The self-conditioning of (Chen et al., 2022b; Yuan et al., 2022) is also incorporated into the denoising function $\mathsf{ED}_\theta^0(\mathbf{y}_t, \mathbf{x}, t)$, in a manner that the former prediction $\tilde{\mathbf{y}}_t$ is further concatenated with the noisy sample $\mathbf{y}_t$, as presented in Appendix B.

We train the denoising function $\mathsf{ED}_\theta^0(\cdot)$ and embedding parameters using the training objective of (Yuan et al., 2022), as presented in Appendix C.

## 3.2 N-gram-based Generative Retrieval

The remaining part is to apply $\mathsf{Gen}_{ngram}(\cdot)$ and $\mathsf{R}_\mathcal{Q}(\cdot)$ for retrieving an original document from the sampled pseudo-document $\mathbf{q} \sim \mathsf{Diffu}(\mathbf{x})$. Following (Bevilacqua et al., 2022), we first train a sequence-to-sequence model to generate n-grams on FM-indices of (Ferragina and Manzini, 2000), referred to as $\mathsf{Gen}_{ngram}(\cdot)$. Suppose that $\mathcal{Q}$ is a set of n-grams resulting from $\mathsf{Gen}_{ngram}(\mathbf{q})$, where each n-gram occurs at least once in the given collection $\mathcal{C}$. Then, we compute $\mathsf{R}_\mathcal{Q}(d, \mathbf{q})$, the relevance score of a document $d \in \mathcal{C}$ by checking how $d$ is matched well with the generated n-grams $\mathcal{Q}$.

To formally describe $\mathsf{R}_\mathcal{Q}(d, \mathbf{q})$ as in (Bevilacqua et al., 2022), for each fixed-length n-gram $g \in \mathcal{Q}$ with $|g| = k$, we first compute its normalized frequency from the FM-index as follows:

$$
P(g) = \frac{c(g, \mathcal{C})}{\sum_{d \in \mathcal{C}} |d|}
\tag{5}
$$

where $c(g, \mathcal{C})$ is the frequency of the n-gram $g$ in the collection and $|d|$ is the length of document

[5]. Now, let $P(g|\mathbf{q})$ be the generative probability of $g$ obtained by the decoder. The n-gram weight function is defined as follows (Bevilacqua et al., 2022):

$$
w(g, \mathbf{q}) = max\left(0, \log \frac{P(g|\mathbf{q})(1 - P(g))}{P(g)(1 - P(g|\mathbf{q}))}\right)
$$

For each document $d \in \mathcal{C}$, we obtain $\mathcal{Q}^{(d)} \subseteq \mathcal{Q}$ a set of the n-grams with the highest weights that do not overlap each other in the document, so that maximizes $\sum_{g \in \mathcal{Q}^{(d)}} w(g, \mathbf{q})$.

Finally, the relevance score of $d$ is the weighted sum of all n-grams in $\mathcal{Q}^{(d)}$ as follows:

$$
\begin{aligned}
&R_\mathcal{Q}(d, \mathbf{q}) \\
&= \sum_{g \in \mathcal{Q}^{(d)}} w(g, \mathbf{q})^\gamma \left(1 - \kappa \frac{|\text{set}(g) \cap C(g)|}{|\text{set}(g)|}\right)
\end{aligned}
\tag{6}
$$

where $\gamma, \kappa$ are hyperparameters, $\text{set}(g)$ is the set of tokens in $g$, and $C(g)$ is the coverage set, the union of all other n-grams in $\mathcal{Q}$ with a higher score, defined as follows:

$$
C(g) = \bigcup_{g' \in \mathcal{Q}, p(g'|\mathbf{q}) > p(g|\mathbf{q})} set(g')
\tag{7}
$$

Details of the contrained decoding for $\mathsf{Gen}_{ngram}(\mathbf{q})$ using the FM-index are present in Appendix D. It is also worth noting that SEAL (Bevilacqua et al., 2022) uses uniformly sampled spans from documents in the collection as "unsupervised" samples to train, aiming to expose the model to more pieces of evidence. In our setting, the generated documents contain semantically-rich contextual information; thus, we do not require such additional data.

## 4 Experiments

### 4.1 Datasets

We conducted experiments using two datasets: MS MARCO (Nguyen et al., 2016) and Natural Questions (Kwiatkowski et al., 2019).

**MS MARCO.** MS MARCO dataset was collected from Bing search queries and the corresponding web page documents. For the passage retrieval task, the full collection contains approximately 500k query-document pairs. For a fair comparison with previous works, we sample the 100k subset following (Zhuang et al., 2022; Ren et al., 2023) for

---

[5] Here, $|d|$ is the number of the occurrence of all fixed length n-grams in $d$, which makes Eq. (5) to sum to 1 over all n-grams.

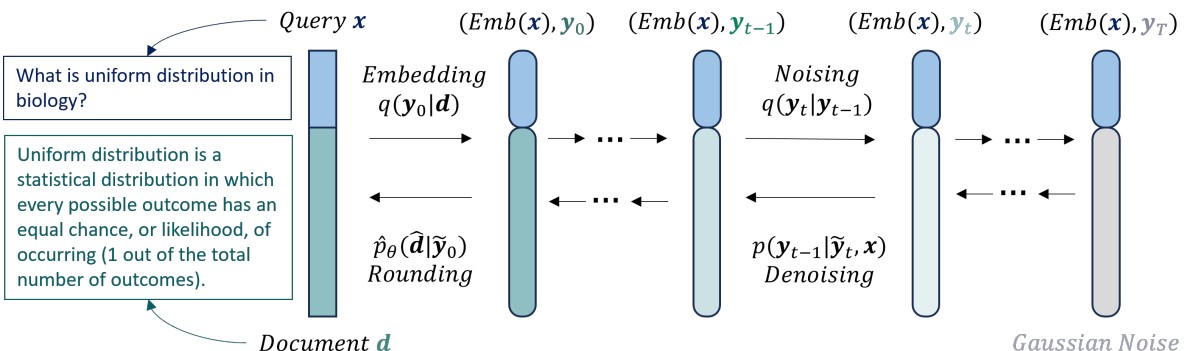

Figure 2: The overall processes involved in diffusion-based document generation, utilizing a sequence-to-sequence approach: i) the forward process first applies the embedding function $\mathsf{Emd}(\cdot)$ to convert a discrete sequence to continuous vectors and then follows the diffusion process of using $q\left(\mathbf{y}_t|\mathbf{y}_{t-1}\right) = \mathcal{N}\left(\mathbf{y}_t; \sqrt{\alpha_t}\mathbf{y}_{t-1}, (1-\alpha_t)\mathbf{I}\right)$ to gradually add Gaussian noise during the remaining time steps; ii) The reverse process performs the denoising function $\mathsf{ED}_\theta^0\left(\mathbf{y}_t, \mathbf{x}, t\right)$ under $p_\theta\left(\mathbf{y}_{t-1}|\mathbf{y}_t, \mathbf{x}\right)$ of Eq. (3) during time steps until $t=0$ and finally applies the rounding function $\mathsf{Round}(\cdot)$, converting continuous vectors to a discrete sequence, as summarized in Eq. (4).

training, referred to as **MS MARCO 100k**. For evaluation, we use the "full dev" set containing 6980 query-document pair samples. We also test our model on a 300k subset to explore the zero-shot setting of the diffusion model.

**Natural Questions.** Natural Questions dataset was collected from natural language questions and Wikipedia pages using Google. We use the processed dataset following NCI (Wang et al., 2022) and SEAL (Bevilacqua et al., 2022), which containing approximately 307k query-document training pairs. For evaluation, we also use the "full dev" set for evaluation containing 7830 test samples, referred to as **NQ 320k**, as in (Tay et al., 2022).

### 4.2 Evaluation Metrics

We report the Hits@1 and Hits@10 metrics, indicating the retrieval success rates of the top-1 and top-10 documents in the ranking list among all evaluation samples, respectively.

### 4.3 Baseline

We selected a subset of prevalent sparse/dense retrieval methods and a set of of the recently proposed generative retrieval methods as baselines for comparison.

For sparse retrieval methods:

- **BM25** (Yang et al., 2017) is a classical sparse retrieval model that evaluates the similarity between queries and documents based on TF-IDF for retrieval.
- **docT5query** (Nogueira et al., 2019) is another strong sparse retrieval method based on document expansion that first predicts possible

related queries based on the document, concatenates them to the end of the document, and then performs the BM25 retrieval.

For dense retrieval methods:

- **DPR** (Karpukhin et al., 2020) is the most representative dual-encoder model based on FAISS index (Johnson et al., 2019).
- **ANCE** (Xiong et al., 2021) is a dual-encoder model using hard negative samples for training, based on an approximate nearest neighbor search.

For generative retrieval methods:

- **DSI** (Tay et al., 2022) is a pioneering work on generative retrieval that jointly trains indexing and retrieval tasks based on the T5 (Raffel et al., 2020) model.
- **DSI-QG** (Zhuang et al., 2022), in addition to DSI, generates multiple queries for documents for data augmentation and trains indexing models with those generated queries to tackle the data distribution mismatch in DSI.
- **SE-DSI** (Tang et al., 2023a) is another work based on DSI that uses learning strategies from Cognitive Psychology to induce a semantic-enhanced DSI model.
- **SEAL** (Bevilacqua et al., 2022) is a generative retrieval model generating n-grams of documents from a query to enable retrieval, and our second-stage retrieval model is highly based on this work.
- **NCI** (Wang et al., 2022), referring to a Neural Corpus Indexer, adopts multiple query gen-

| Model | Hits@1 | Hits@10 |
|---|---|---|
| BM25 | 46.50 | 73.90 |
| BM25 + docT5query | 68.50 | 92.70 |
| DPR | 65.00 | 91.90 |
| DSI-base | 2.10 | 7.10 |
| DSI-large | 10.40 | 23.00 |
| SE-DSI | 53.47 | 75.28 |
| DSI-QG-base | 59.20 | 81.10 |
| DSI-QG-large | 58.40 | 80.90 |
| TOME | 71.93 | - |
| DiffusionRet | 85.29 | 96.62 |

Table 1: Retrieval performances (Hits@k) on test sets of MS MARCO 100k, comparing DiffusionRet with other previous baselines.

| Model | Hits@1 | Hits@10 |
|---|---|---|
| BM25 | 29.27 | 60.15 |
| BM25 + docT5query | 39.13 | 69.72 |
| DPR | 50.20 | - |
| ANCE | 52.63 | - |
| DSI-large | 35.60 | 62.60 |
| DSI-XL | 39.10 | 66.80 |
| DSI-XXL | 40.40 | 70.30 |
| DSI-QG-base | 63.49 | 82.36 |
| DSI-QG-large | 65.13 | 82.50 |
| SEAL | 59.93 | 74.50 |
| NCI | 66.23 | - |
| TOME | 66.64 | - |
| GenRet | 68.10 | - |
| DiffusionRet | 71.66 | 80.23 |

Table 2: Retrieval performances (Hits@k) on test sets of NQ 320k, comparing DiffusionRet with other previous baselines.

eration strategies and a prefix-aware weight-adaptive decoder to improve the retrieval model.

- **TOME** (Ren et al., 2023) is a two-stage generative retrieval method partially similar to our architecture, but it uses the T5 (Raffel et al., 2020) model to generate documents, and the generated documents are used in another T5 model to generate possible web URLs, with the URL as the document identifier.
- **GenRet** (Sun et al., 2023) is a generative retrieval method that learns to tokenize documents into short discrete representations to define document identifiers better using a discrete auto-encoding approach.

For comparison, we present the results reported in the original paper as baseline performance. When multiple same-method performances are mentioned in other studies, we report the best performance.

### 4.4 Implementation Details

**Data Augmentation.** To train DiffusionRet, we applied data augmentation by directly performing query generation or adopting existing generated queries. For the MS MARCO 100k dataset, we used the model in the docT5query (Nogueira et al., 2019) pre-trained on the MS MARCO dataset, to generate 5 corresponding queries for each training document. For the NQ 320k dataset, we used the same dataset as previous studies (Wang et al., 2022; Bevilacqua et al., 2022) where 10 queries were generated for each training document [6].

---

[6] NCI (Wang et al., 2022) made their processing scripts and the processed dataset publicly available

**Training Diffusion Model.** To train the diffusion model for document generation in Section 3.1, we followed the work of SeqDiffuSeq (Yuan et al., 2022) using a 12 layers encoder-decoder Transformer (Vaswani et al., 2017) model as the backbone. We set the diffusion steps to 2,000, as in the most previous works (Li et al., 2022b; Gong et al., 2023; Yuan et al., 2022). Some generated examples using the diffusion model are presented in Appendix A.

**Training Retrieval Model.** To train the generative retrieval model in Section 3.2, SEAL (Bevilacqua et al., 2022) was adopted but under modified settings for our case. In our training dataset, we built a set of pairs of (*pseudo-document*, *n-gram*), where a generated pseudo-document is a source sequence, while target n-grams are sampled from the original "ground truth" document, not from the generated one. Beyond the use of short n-grams, we gathered 10, 30-long n-grams from the original document using SEAL's sample strategy. For each pseudo-document, we sampled 10 target n-grams of sequence length 30 from th e corresponding ground truth document using SEAL's sample strategy.

For the MS MARCO 100k dataset, we performed a sampling process of the diffusion model on the original 100k queries, obtained a set of 100k generated pseudo-documents, and finetuned the BART-large model (Lewis et al., 2019). For the NQ 320k dataset, we sampled pseudo-documents from the original 307k queries for training data and

| variants | | MS MARCO 100k | | NQ 320k | |
|---|---|---|---|---|---|
| query | gen-doc | Hits@1 | Hits@10 | Hits@1 | Hits@10 |
| ✓ | | 40.47 | 71.15 | 39.11 | 61.95 |
| | ✓ | 80.69 | 93.35 | 66.07 | 73.40 |
| ✓ | ✓ | 85.29 | 96.62 | 71.66 | 80.23 |

Table 3: Ablation study for DiffusionLM on MS MARCO 100k and NQ 320k where "query" and "gen-doc" refer to the runs where a final query includes an original query $\mathbf{x}$ and a generated document $\mathbf{q}$, respectively.

| Hits@1 | Hits@2 | Hits@3 | Hits@5 | Hits@10 |
|---|---|---|---|---|
| 65.44 | 73.42 | 78.35 | 83.26 | 87.94 |

Table 4: Retrieval performances (Hits@k) on MS MARCO 300k (with 200k zero-shot data), under the diffusion model trained from the MS MARCO 100k as in Table 1.

finetuned the model from SEAL's NQ checkpoint, where other parameter settings remain the same as those of (Bevilacqua et al., 2022).

**Inference.** Given a query, DiffusionRet first applies the diffusion model to generate a pseudo-document and then performs constrained decoding of the generative retrieval model to generate a set of long n-grams, as in Eq. (1). The FM-index was built on the union of the training and dev sets to restrict the decoding search space, as in SEAL (Bevilacqua et al., 2022). During the constrained decoding, we prepend an original query to a pseudo-document to improve performance.

More detailed settings are presented in Appendix H.

## 5 Results and Analysis

### 5.1 Main Results

**Results on MS MARCO 100k.** Table 1 lists the comparison results on MS MARCO 100k. DiffusionRet significantly outperforms all other baseline methods for sparse/dense retrieval and generative retrieval.

**Results on NQ 320k.** Table 2 shows the comparison results on the NQ 320k dataset. Again, DiffusionRet method achieves superior results on Hits@1 metric over all baseline methods in previous works. For Hits@10, DiffusionRet substantially outperforms most generative retrieval methods, including SEAL, but underperforms compared to DSI-QG (Zhuang et al., 2022). The relatively weak performance on the NQ 320k dataset compared to the case of MS MARCO 100k is possibly because the diffusion model may require a large

parameter size for document generation as the number of documents in the collection increases. In this study, we use a vanilla Transformer backbone with few parameters, which might be insufficient for generating longer documents from a shorter query, particularly when the collection size is large, compared with other simple generation tasks (such as question generation and text simplification). A valuable future direction would be to handle low-quality pseudo-documents by improving the current settings of DiffusionRet.

### 5.2 Ablation Study

In this section, we conduct ablation studies to analyze the DiffusionRet strategies.

Table 3 examines the effects of removing either an original query or a generated document when formalizing an extended query for the generated retrieval. "Query" indicates whether an original query $\mathbf{x}$ is prepended to a generated document $\mathbf{q}$; and "gen-doc" whether a generated document is provided for forming a final query for the n-gram generation.

As Table 3 shows, the runs using a generated document (i.e., the 2nd-3rd rows) lead to significant improvements compared to that with only an original query (i.e., the 1st row). When using only an original query, the performances are largely poor on both datasets. The results confirm that the setting of DiffusionRet using both the generated document and original query is effective.

### 5.3 Expansion of Diffusion Model to Unseen Documents on MS MARCO 300K

We further evaluate whether DiffusionRet's diffusion model is robust for other unseen documents. To extend the document space, we take a subset of 300k pairs, called **MS MARCO 300K**, i.e., the number of documents is 300k. We use the current diffusion model trained using 100k examples and apply it to the remaining 200k unseen queries to generate 200k pseudo-documents. We use these 300k data to train the retrieval model and build

| Model | source | Hits@1 | Hits@10 | Params |
|---|---|---|---|---|
| - | query | 40.47 | 71.15 | 0 |
| T5-base | gen-doc | 19.53 | 45.63 | 220M |
| T5-base | query + gen-doc | 45.79 | 76.05 | 220M |
| Diffusion | gen-doc | 80.69 | 93.35 | 103M |
| Diffusion | query + gen-doc | 85.29 | 96.62 | 103M |

Table 5: Comparison of retrieval performances between diffusion models with SLMs on MS MARCO 100k, where the T5-base model is used for SLM; "query", "gen-doc", and "query + gen-doc" indicates the SEAL's runs whose input is an orginal query, a generated pseudo-document, and their concatenated sequence, respectively.

| Model | Params |
|---|---|
| DSI-large | 800M |
| DSI-XL | 3B |
| DSI-QG-base | 800M + 336M + 220M |
| DSI-QG-large | 800M + 336M + 800M |
| TOME | 800M + 220M |
| DiffusionRet | 103M + 406M |

Table 6: Comparison of model parameter sizes between DiffusionRet, TOME, DSI and DSI-QG.

the FM-index decoding space. Table 4 shows the results, indicating that DiffusionRet, even in an enlarged document search space, yields the best results in terms of Hits@10 and outperforms the other baseline models in Hits@1, except for TOME.

### 5.4 Comparison of Diffusion Models with Small Language Models

To further examine whether the diffusion model improves small language models (SLM) in generating pseudo-documents, we select a T5-base (Raffel et al., 2020) model as a SLM, which has roughly twice the number of parameters (220M) compared to the diffusion model, and train it for the first document generation stage on MS MARCO 100k. Table 5 shows the comparison result in retrieval performances on MS MARCO 100k between methods using the diffusion and T5-base models. It is shown that the diffusion model significantly outperforms the T5-base model, confirming that diffusion models are able to generate high-quality pseudo-documents in a parameter-efficient manner. It should be noted that The T5-base model is not effective in generating pseudo-documents, being failed to improve the baseline in the setting using the generated documents aglone (i.e., gen-doc), whereas the diffusion model consistently achieves marked improvements irrespective of whether an original query is appended or not.

### 5.5 Comparison of Model Parameter Size

To further demonstrate the parameter efficiency of DiffusionRet, Table 6 summarizes the model parameter sizes, comparing with those in related works. DiffusionRet uses a 12-layer Transformer model (Vaswani et al., 2017) with 103M parameters for the diffusion model backbone and a BART-large model (Lewis et al., 2019) with 406M parameters as the generative retrieval model. As listed in Table 6, DiffusionRet has the smallest parameter size and reduces the required size of TOME (Ren et al., 2023) consisting of a document generation model based on T5-large and a URL generation model based on T5-base.

## 6 Conclusion

In this paper, we proposed DiffusionRet, a two-stage architecture consisting of diffusion-based document generation and n-gram-based generative retrieval. Extensive experiments on the MS MARCO and NQ datasets showed that DiffusionRet achieved state-of-the-art performances in terms of Hits@1, outperforming most existing retrieval models. Further analysis confirmed that the use of generated documents as a final query was the key component to bring about those significant improvements.

In future work, we would like to invent a full-fledged diffusion model to jointly generate and retrieve without requiring a subsequent retrieval component. Furthermore, we would also jointly perform a document generation and retrieval in a single transformer model by viewing document generation as the chain-of-thought (Wei et al., 2022). We would simplify refining n-gram-based generative retrieval such that the generation and retrieval components are less separated. Comparing the full-fledged diffusion-based generate-then-retrieval model with popular LLMs would be interesting and inspire future research.

## Limitations

Although the diffusion model has the advantage of few parameters, it performs thousands of diffusion steps to sample the data during inference, slowing the inference speed. Although the adopted work (Yuan et al., 2022) considerably improved the inference speed, it is still non-trivial for us to perform extensive sampling of the original training set to construct the training data for our retrieval model. Incorporating some studies including the consistency model (Song et al., 2023) based on ordinary differential equations (ODEs) which can significantly accelerate the inference would be a valuable future work.

Another limitation is the two-stage style, performing generation and retrieval in a cascaded manner. While generative retrieval provides an elegant framework that integrates indexing and retrieval into the transformer's parameters, DiffusionRet uses two different types of models for generation and retrieval. Thus, an arguably desirable approach is to unify the two stages in a single diffusion model or a unified transformer without losing the principled and elegant manner of generative retrieval.

In addition, our current experiments show the major performances without analyzing the parts with more errors, helping us to move a step towards promising future directions. As we establish two stage components, it is possible to evaluate the generation and retrieval modules separately, which is worthy of future research. In particular, we need to explore the relationship between the parameter sizes of diffusion model and collection size. This is helpful in determining the situation required to increase the parameter size of the diffusion model.

## Acknowledgements

This work was partly supported by the Institute of Information & communications Technology Planning & Evaluation (IITP) grant funded by the Korea government (MSIT) (No. RS-2022-00187238, Development of Large Korean Language Model Technology for Efficient Pre-training). Shanbao Qiao and Xuebing Liu were also supported by China Scholarship Council (CSC).

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

## A  Examples of Generated Documents

We show some examples of diffusion models generated documents by using different color fonts to mark different quality of the generated content.
**Green:** correct and relevant content.
**Yellow:** relevant content in similar domain or imprecise knowledge.
**Red:** irrelevant content or incorrect knowledge.

For seen data (in training set):
**Query:** *who plays mantis in guardians of the galaxy*
**Ground truth document:**
*pom klementieff pom klementieff ( born 3 may 1986 ) french actress. she trained cours florent drama school paris appeared films loup ( 2009 ) sleepless night ( 2011 ) hacker's game ( 2015 ). she plays role mantis film...*
**Generated document:**
*pom klementieff pom klementieff ( born c. 1986 / 1987 ) french actress. she trained coeg florent drama school paris appeared films loup ( 2009 ) mouthless night ( 2011 ) book 's game ( 2015 ). she plays role man...*

For unseen data (in dev/eval set):
**Query:** *what are chia seeds benefits*
**Ground truth document:**
*health benefits of chia seeds. chia seeds impart many health benefits, some of which include controlling diabetes, lowering cholesterol levels, improving the digestive system, boosting heart health, controlling hypertension, and they even contain anti - inflammatory properties.*
***Generated document:***

*chia seeds are a good source of protein. the seeds are rich in dietary plants, wheat, fiber, and dental fiber. fruits so eat a and to the digest digestiveive system, potassium, fruits, and vegetables products. all chia are rich in fruit and developing and have and to health and properties.*

There are also examples of failed generation, usually for long-tail data or where the query is not very relevant to the document, such as the following example. And there would be a future work to handle these low-quality samples.

**Query:** *aarra definition government*

**Ground truth document:**

*american recovery reinvestment act 2009 the american recovery reinvestment act 2009 ( arra ) ( pub. l. 111 - - 5 ) nicknamed recovery act stimulus package enacted 111th u. s. congress signed law president barack obama february 2009. developed response great recession arra's primary objective ...*

**Generated document:**

*federal government - the government federal government government community government 1. 1. 1. 1. 1. 1. 1. 1. 1. 1. 1. 1. 1. 1. 1. 1. 1. 1. 1. 1. 1. 1. 1. 1. 1.*

We also provide some document samples generated by T5-base as comparison.

**Query:** *visable light from the sun*

**Ground truth document:**

*The incoming energy from the Sun to Earth is mainly visible sunlight, called the visible portion of the spectrum of electromagnetic radiation. We perceive visible sunlight as colors from violet (short-wave radiation) to red (long-wave radiation) ...*

**Generated document:**

*The visible sun's light is the most visible light from the Earth, and that is why it is the cheapest light source. It is a*

**Query:** *define eurasia*

**Ground truth document:**

*Define Eurasia . Eurasia. Eurasia is of major geographic and geologic significance. It is also important to the history of human civilization. Answer and Explanation: Eurasia is a landmass that consists of the continents of Asia and Europe. It measures over 21 million square miles ...*

**Generated document:**

*European Union (Europe): An independent region of the continent of Eastern Europe. An independent, independent nation or state in the Eurasia. The Union*

## B  Denoising Process of Diffusion-based Document Generation

To describe $A_t$ and $B_t$ in Eq. (3), we follow additional notation of $\bar{\alpha}$ of (Yuan et al., 2022), as follows:

$$\bar{\alpha} = \prod_{s=1}^{t} \alpha_s$$
$$\beta_t = 1 - \alpha_t \tag{8}$$

$A_t$ and $B_t$ are defined as follows:

$$A_t = \frac{\sqrt{\bar{\alpha}_{t-1}}\beta_t}{1 - \bar{\alpha}_t}$$
$$B_t = \frac{\sqrt{\alpha_t}(1 - \bar{\alpha}_{t-1})}{1 - \bar{\alpha}_t} \tag{9}$$

According to Eq. (3), $\mathbf{y}_{t-1}$ is sampled by mixing the formed predicted sequence $\tilde{\mathbf{y}}_0^t = \mathsf{ED}_\theta^0(\mathbf{y}_t, \mathbf{x}, t)$, and $\mathbf{y}_t$, and random noise.

The self-conditioning (Chen et al., 2022b) additionally takes the formed predicted sequence $\tilde{\mathbf{y}}_0^t$ to predict the current sequence $\tilde{\mathbf{y}}_0^{t-1}$, using the extended denoising function $\mathsf{ED}_\theta^0(\mathbf{y}_t, \tilde{\mathbf{y}}_0^t, \mathbf{x}, t)$. The encoder part takes $\mathbf{x}$, while the decoder part takes the concatenated sequence of $\mathbf{y}_t$ and $\tilde{\mathbf{y}}_0^t$ with the dimensionality of $n \times 2d$.

## C  Training objective of Diffusion model

We optimized the denoising parameter $\theta$ in $\mathsf{ED}_\theta^0(\cdot)$ and the embedding parameters by minimizing the variational lower bound of data log-likelihood. Following the derivation of (Yuan et al., 2022), when a pair of a query $\mathbf{x}$ and a document $\mathbf{d} \in \mathbb{R}^{n \times |\mathcal{V}|}$ is given, the training objective is simplified to:

$$\mathbb{E}_{q(\mathbf{y}_0, \mathbf{x}, \mathbf{d})} \left[ \sum_{t=2}^{T} \mathbb{E}_{q(\mathbf{y}_t | \mathbf{y}_0)} \left\| \mathsf{ED}_\theta^0(\mathbf{y}_t, \mathbf{x}, t) - \mathbf{y}_0 \right\|^2 \right.$$
$$+ \left\| \mathsf{ED}_\theta^0(\mathbf{y}_1, \mathbf{x}, 1) - \mathsf{Emd}(\mathbf{d}) \right\|^2$$
$$\left. + \left\| \sqrt{\bar{\alpha}_T} \mathbf{y}_0 \right\|^2 + \log p(\mathbf{d} | \mathbf{y}_0) \right]$$

## D  Constrained Decoding using FM-Index

To meet the requirements that all the generated sequences appear in the corpus and that the location

of the sequences can be identified for retrieval, the FM-index (Ferragina and Manzini, 2000) is a suitable efficient data structure that can obtain all the possible successor tokens for constrained decoding in a desired time ($\mathcal{O}(|V|\log|V|)$). Where $V$ is the vocabulary of the model, hence the computational time complexity is independent of the corpus size. In addition, because the FM-index is a self-index in the form of a compressed suffix array, it does not require a large storage space and is usually smaller than the corpus data under the vocabulary size of regular language models.

The FM-index is built based on the Burrows-Wheeler Transform (BWT, Burrows, 1994), where all the rotations of the string are listed in a matrix and sorted lexicographically. Given an example string "APPLE," where $ denotes the end token, the corresponding matrix M is shown in Figure 3. The L column is the BWT of the string, and the FM-index only stores the F and L columns to achieve an efficient localization of an arbitrary string by performing a reverse lookup of the token's prefix in F at the same location in L, and then iteratively extending to any n-gram in the corpus.

Figure 3: BWT Matrix

When applied to the autoregressive model, the FM-index is used to constrain the decoding space within a possible next token set, and the logit of all unconsidered tokens (out of the set) is masked to $\infty$. When iteratively generating the target sequence, the next new token is selected from the ranks in L in the same row as the previous token in column F. Thus, the beam search can return a set of n-grams appearing in the corpus and the location of the n-grams can be easily obtained using the FM-index row number. After decoding, we also used the FM scoring function to better select n-grams and intersective scoring functions to better rank the retrieved documents following SEAL's (Bevilacqua et al., 2022) strategy.

# E   Datasets Information

Microsoft machine reading comprehension (MS MARCO) (Nguyen et al., 2016) is a collection of datasets focused on deep learning in search and contains queries of real Bing searches and corresponding document answers. We used the MS MARCO passage corpus provided by Tevatron (Gao et al., 2022a) containing approximately 500k query-document pairs for training. Following previous work that used different scales of data (Tay et al., 2022; Zhuang et al., 2022; Ren et al., 2023), we downsampled a 100k subset for training and used the full dev set for evaluation containing 6980 query samples.

Natural Questions (NQ) comprises a collection of natural language questions and Wikipedia page documents. The training set contains 307k query-document pairs, and the dev set has 7830 queries, called the NQ 320k in DSI (Tay et al., 2022). We used the full training and dev sets in the experiments.

# F   Rouge-Score of Generated Documents

We calculated the Rouge-score between the generated document and ground truth document on MS MARCO, and the data in the training set are seen for the model, while those in the dev set are unseen for the model. Table 7 shows the results.

| Set | Rouge-1 | Rouge-2 | Rouge-L |
|-----|---------|---------|---------|
| train | 0.696 | 0.537 | 0.677 |
| dev | 0.322 | 0.123 | 0.257 |

Table 7: Rouge-score of Generated Documents.

Despite the low Rouge-score on dev set, the generated documents still offer rich keywords and valuable semantic information for retrieval.

# G   Detailed Hits@k

Figure 4-5 presents detailed Hits@k results, with varying k, in comparison to the best performing baselines on MS MARCO 100k and NQ 320k, respectively.

# H   Detailed Settings

**Settings for the diffusion model used in document generation**: For training examples in the

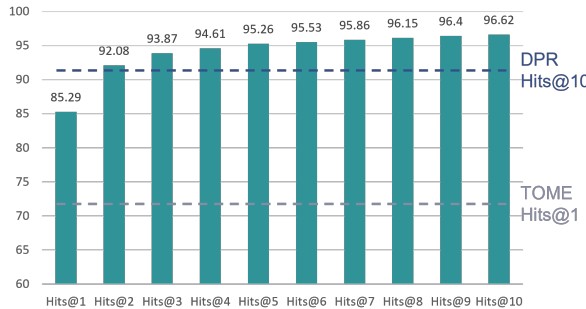

Figure 4: Hits@k on MS MARCO 100k, varying the number of retrieved documents (k). The results are compared with TOME (Ren et al., 2023) for Hits@1 and DPR (Karpukhin et al., 2020) for Hits@10 metric.

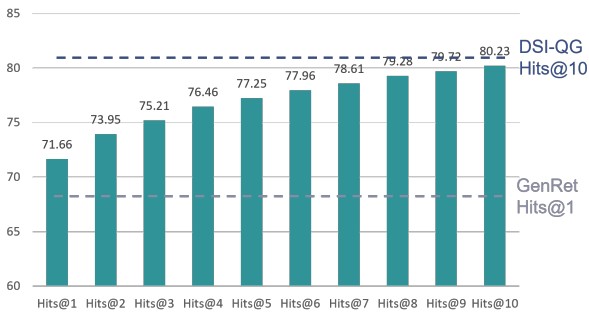

Figure 5: Hits@k on NQ 320k varying the number of retrieved documents (k). The results are compared with GenRet Sun et al., 2023 for Hits@1 and DSI-QG, Zhuang et al., 2022 for Hits@10.

MS MARCO and NQ datasets, we used pairs of query-document where all queries were automatically generated for each document.

The maximum length of the input query-document pair was capped at 128 because the vast majority of sequence lengths fell within this range, and the maximum length of the output was set to 64. We implemented the sequence-to-sequence diffusion model using PyTorch, and trained the model on 4 Nvidia TITAN-RTX GPUs. The optimizer used was AdamW. The detailed hyperparameter settings are as follows:

- Optimizer: AdamW

- Learning rate: 1e-4

- Maximum input length: 128

- Maximum output length: 64

- Diffusion steps: 2000

- Batch size: 192

- Warmup steps: 10000

- Total training steps: 500k / 1M for MS MARCO 100k / NQ 320k dataset

**Settings for the generative retrieval model**:

The generative retrieval model was based on SEAL's work (Bevilacqua et al., 2022), and the model was fine-tuned from the BART-large model (Lewis et al., 2019) on 4 Nvidia RTX 8000 GPUs. The training was performed using Fairseq Toolkit. The detailed hyperparameter settings are as follow:

- Optimizer: Adam

- Learning rate: 3e-5

- Base model: BART-large

- Tokens per GPU: 4096

- Warmup steps: 500

- Hyperparameter $\gamma$: 2.0

- Hyperparameter $\kappa$: 0.8

- Total training epochs: 96 / 18 for MS MARCO 100k / NQ 320k dataset

Specifically, for MS MARCO, we ran the training for 96 epochs with a learning rate of 3e-5 because the loss could not be further optimized. For the NQ 320k dataset, we fine-tuned SEAL's NQ check-point during 18 epochs when the loss reached stability, while other parameter settings remained the same as (Bevilacqua et al., 2022). We did not use the uniformly sampled document spans as "unsupervised" samples to expose evidence during training, different from SEAL (Bevilacqua et al., 2022), since our generated documents already have semantically-rich contextual information.