# OpenReview forum: "DiffusionRet: Diffusion-Enhanced Generative Retriever using Constrained Decoding"
_EMNLP/2023/Conference — EMNLP 2023 Findings_

### Official Review · Reviewer_qdE4 · 2023-08-04

**Soundness:** 3

**Excitement:**

3: Ambivalent: It has merits (e.g., it reports state-of-the-art results, the idea is nice), but there are key weaknesses (e.g., it describes incremental work), and it can significantly benefit from another round of revision. However, I won't object to accepting it if my co-reviewers champion it.

**Missing References:**

The following paper considered generating synthetic identifiers, which are longer n-grams.

[3] Li, Y., Yang, N., Wang, L., Wei, F., & Li, W. (2023). Multiview Identifiers Enhanced Generative Retrieval. ArXiv, abs/2305.16675.

**Paper Topic And Main Contributions:**

This paper aims to propose a new generative retrieval method, DiffusionRet. It first expands a given query with a pseudo document generated by a diffusion model. Then it generates long n-grams that would be mapped to an original document in the collection, given the original query and pseudo document. The experimental results show that DiffusionRet not only exhibits great performance but also has a smaller number of parameters compared to strong baselines.



**Questions For The Authors:**

Question A:
The authors claim, "For MS MARCO 100k dataset, we apply the inference based on the diffusion model on the original 100k queries, obtain a set of 100k generated pseudo documents ... For NQ 320k dataset,  we sample pseudo documents from the original 307k queries". I was wondering why the sampling is only for NQ 320k.

Question B:
The caption of Table 1, "Retrieval performances (Hits@k) on test sets of MS MARCO 100k, comparing DiffusionRet with other previous baselines". What are the test sets of MS MARCO 100k? Is the full dev set (6980 queries) a test set? If the full dev set is regarded as a test set, there would be no development set. If so, I would like to know how the authors tune the hyperparameters without a development set.


**Reasons To Accept:**

1. Conduct query expansion by a diffusion-based document generation method.
2. Consider generative retrieval with long n-grams (previous studies used much shorter n-grams).
3. The experiments show the effectiveness of DiffusionRet.

**Reasons To Reject:**

1. One important claim in the introduction is not supported by experiments: the authors claim, "Different from the setting in [1] where the length of n-grams is small (i.e., 3-to-5), we consider long n-grams with the length of 30, which are more likely unique to a specific document". However, there is no corresponding analysis (e.g., a variant of DiffusionRet generating shorter n-grams) in the paper and so it's unclear if generating longer n-grams could really bring about improvements. The length of n-grams is the only difference between this paper and [1] in terms of the phase of retrieval.
2. The idea of feeding the original query and pseudo document into a generative retrieval method is really similar to query expansion (e.g., [2]). However, the difference from query expansion is not discussed in the paper.

[1] Bevilacqua, M. ,  Ottaviano, G. ,  Lewis, P. ,  Yih, W. T. ,  Riedel, S. , &  Petroni, F. . (2022). Autoregressive search engines: generating substrings as document identifiers.
[2] Wang, L., Yang, N., & Wei, F. (2023). Query2doc: Query Expansion with Large Language Models. ArXiv, abs/2303.07678.

**Reproducibility:**

4: Could mostly reproduce the results, but there may be some variation because of sample variance or minor variations in their interpretation of the protocol or method.

**Reviewer Confidence:**

3: Pretty sure, but there's a chance I missed something. Although I have a good feel for this area in general, I did not carefully check the paper's details, e.g., the math, experimental design, or novelty.

---

> ### Author Rebuttal · Authors · 2023-08-29
>
> ## Response to Reviewer qdE4
> Thank you for raising valuable discussion and questions! We appreciate that the reviewer mentioned the good performance and smaller number of parameters of our work compared to strong baselines. In this discussion we hope that we can further address all your concerns as well as answer the questions.
>
> ### 1. Response to Reason to Reject 1: Discussion about the N-gram length and other difference between DiffusionRet and SEAL.
>
> We are sorry for making this confusion. While it is true that the length of n-gram in our n-gram generative retrieval is different from SEAL [1], it is hard to say that this is our major claim. Please understand that we clarify the difference but we do not claim here that the just use of long lengths of n-gram lead to the improvement in the retrieval performance. More precisely speaking, by putting the first stage for generating a pseudo-document, because its content and length are similar to those of a document (rather than a short query), it is now possible to use large n-grams under the SEAL framework, less likely causing the “unmatched” situation with long n-grams.
>
> Although we don’t fully explore the advantage of using long n-grams, we believe that it can improve the retrieval speed, as the number of required n-gram terms keep much small; at the extreme case, we can only keep single 30-length long ngram, which will likely significantly reduce the retrieval time of using multiple short terms.
>
> In addition, as replied to the reviewer PVJR, the difference between SEAL [1] and our n-gram generative retrieval is not only the length of n-gram, but also the requirement of the “extensive data augmentation” like query expansion. In the original SEAL [1], spans from a document are uniformly extracted and they are used as additional queries, as much like the data augmentation using the query generation similar to DSI-QG [2] and NCI [3]. On the other hand, our n-gram generative retrieval does not require such data augmentation, where only a single pseudo document per document is used as the source part (not multiple ones), where n-grams are used as target part; detaching such data augmentation from SEAL framework becomes possible resulting from the two-stage setting by putting explicitly the pseudo document generation part. As a result, it is expected that our n-gram generative retrieval can be much efficiently trained comparing to the original SEAL, given the manner that does not rely on such extensive data augmentation.
>
> We are sorry again for this confusion. If given a chance, we will definitely revise our paper such that the difference in length is not our major and key claim, in any revised version of the paper.
>
> Also, to clarify the difference between our method and SEAL, (as similarly replied to the reviewer PVJR), if given a chance, we would like to add the single-stage SEAL equipped with the data augmentation in our own experiment settings on MS MARCO 100k and NQ 320k, and revise the main text and appendix such that the training difference and other important ones are clearly revealed, etc, in any extended version of the paper.
>
>
> ### 2. Response to Reason to Reject 2: Discussion about the differences between DiffusionRet and query2doc.
>
> Thank you for introducing the relevant work. We are sorry for missing this important work. As in query2doc [4], we also believe that LLMs can be applied to generate pseudo documents. But, as similarly replied to the reviewer PVJR, the reasons for using the diffusion model in our work are summarized to 1) there is the recently emerged interest and success in diffusion model and it is interesting and valuable to explore the diffusion model on our addressed semantic inpainting task, 2) the diffusion model is much more parameter-efficient than LLMs; under SLM, the generated results are not satisfactory comparing to that of smaller diffusion model, 3) the semantic inpainting task does not require to generate grammatically perfect sentences, which are one of the key powerful capability in LLMs, so the use of LLMs is not very mandatory, etc.
>
> In addition to the responses on the reviewer PVJR, nowadays LLMs still have some known limitations such as hallucination, and so it is yet still valuable to explore other directions like diffusion model in completing the generation task. Also, diffusion models may different pseudo different samples from those of LLMs, and these samples may be complementary each other; Thus, the use of ensembles or combining multiple evidences from different pseudo documents from diffusion models and LLMs will make promising and interesting direction.
>
> We would like to mention the further difference about query2doc [4].
>
> 1. Parameter size: the most significant difference is the scale of the model. Query2doc [4] employs LLMs as the pseudo-document generator and achieves good results with a 175B LLM, while we only use a diffusion model with 103M parameter as the pseudo-document generator.
> 2. Dependency on parameter size: It is also worth noting that Query2doc presented in the paper that for generative models, the number of parameters has a significant effect on the performance. As they mention in Introduction:
> > _Further analysis also reveals the importance of model scales: query2doc works best when combined with the most capable LLMs while small language models only provide marginal improvements over baselines._
>
>    While our work happens to address the generator's dependence on a large number of parameters using a diffusion model, this was also inspired by the success of previous diffusion model works on controllable text generation and sequence-to-sequence tasks (e.g., Diffusion-LM[5], DiffuSeq[6], SeqDiffuSeq[7], etc., where small diffusion models outperform some of the GPT variants on several different tasks).
>
> 3. Sample quality of diffusion models (in the language perspective): Pseudo-documents generated by the diffusion model may differ from those generated by LLMs. As we mentioned in Section 1, the generation of the diffusion model is viewed as "semantic inpainting" style; When compared with the results of LLMs, the diffusion-generated documents may not be grammatically or semantically perfect sentences (there may be errors in syntax or grammar, etc., as shown in Appendix A), but can provide rich semantic information which is helpful to the next retrieval stage. Relying the second retrieval stage allows us to generate grammatically less-plausible sentences by diffusion model.
>
> Thank you again for letting us the valuable related work. If given a chance, we would definitely add detailed discussion as mentioned above, in any revised version of the paper.
>
> ### 3. Response to Question A: Clarification of "inference" and "sampling"
>
> Thank you for raising this question. We apologize for the ambiguity and would unify the statements in the paper to avoid confusion if given chance. These two statements are essentially the same operation. The inference process of the diffusion model is also known as _sampling_, i.e., sampling step by step from a Gaussian distribution to get the generated result. In our case the inference process is to get the pseudo-document by _sampling_.
>
> ### 4. Response to Question B: About test/dev set and hyper-parameter fine-tune
>
> Thank you for mentioning this issue. We used the full dev set (6980 queries) as test set. This was done on the one hand to maintain the consistency of the evaluation setup with previous related work (e.g., DSI-QG, TOME, etc.), and on the other hand our model achieved the satisfactory performance with the initial hyper-parameter settings (which were set empirically based on previous works), and thus we did not pay much attention to tuning the hyper-parameters.
>
> ### 5. Response to Missing References.
>
> Thank you for recommending the relevant reference. If given a chance, We would happy to include the discussion about this paper in Related Works section. Since this work used different evaluation metrics than ours, it is difficult to add this work as a baseline to the table, fortunately the Hits@5 metrics on NQ dataset was provided, which can be compared to our Hits@5 in Appendix G (MINDER: 65.8, ours: 77.25).
>
> We deeply appreciate your precious comments and suggestions and hope this discussion could address all your concerns. Wish you all the best.
>
>     [1] Bevilacqua, Michele, et al. "Autoregressive search engines: Generating substrings as document identifiers." Advances in Neural Information Processing Systems, 2022.
>     [2] Zhuang, Shengyao, et al. "Bridging the gap between indexing and retrieval for differentiable search index with query generation." arXiv preprint arXiv:2206.10128 (2022).
>     [3] Wang, Yujing, et al. "A neural corpus indexer for document retrieval." Advances in Neural Information Processing Systems 35 (2022): 25600-25614.
>     [4] Wang, Liang, Nan Yang, and Furu Wei. "Query2doc: Query Expansion with Large Language Models." arXiv preprint arXiv:2303.07678 (2023).
>     [5] Li, Xiang, et al. "Diffusion-lm improves controllable text generation." Advances in Neural Information Processing Systems, 2022.
>     [6] Gong, Shansan, et al. "Diffuseq: Sequence to sequence text generation with diffusion models." arXiv preprint arXiv:2210.08933 (2022).
>     [7] Yuan, Hongyi, et al. "Seqdiffuseq: Text diffusion with encoder-decoder transformers." arXiv preprint arXiv:2212.10325 (2022).

---

### Official Review · Reviewer_PVJR · 2023-08-04

**Soundness:** 3

**Excitement:**

3: Ambivalent: It has merits (e.g., it reports state-of-the-art results, the idea is nice), but there are key weaknesses (e.g., it describes incremental work), and it can significantly benefit from another round of revision. However, I won't object to accepting it if my co-reviewers champion it.

**Paper Topic And Main Contributions:**

This paper proposes a two-stage method called diffusion enhanced generative retrieval (DiffusionRet) for generative retrieval. The method consists of two processing steps: first adopting a diffusion model to generate the hypothetical document for the raw query, then preforming N-gram-based generative retrieval like SEAL [1] based on the generated hypothetical document. Experiment results show that DiffusionRet leads to the state-of-the-art performances.

[1] Michele Bevilacqua, Giuseppe Ottaviano, Patrick Lewis, Scott Yih, Sebastian Riedel, Fabio Petroni. Autoregressive Search Engines: Generating Substrings as Document Identifiers.


**Questions For The Authors:**

Q1: How to compare the generation quality and inference speed of a diffusion model of the same size and T5? Lack of some important ablation studies.

**Reasons To Accept:**

1. This paper employs the diffusion model for performing query expansion to generate the hypothetical document as the input of the generative retrieval.

2. The experimental results show that the proposed DiffusionRet performs better on two datasets than previous work. And the ablation study shows the effectiveness of using generated documents.


**Reasons To Reject:**

1. The idea of using the diffusion model to generate the hypothetical document lacks motivation. How would DiffusionRet work with a generative language model (like T5) of the same parameter size?

2. Missing discussion of the difference between the N-gram-based Generative Retrieval used in DiffusionRet and the SEAL model. Is the main difference being the length of ngrams?

3. This paper is poorly formatted with some blank lines (e.g., Page 3 and 8).


**Reproducibility:**

3: Could reproduce the results with some difficulty. The settings of parameters are underspecified or subjectively determined; the training/evaluation data are not widely available.

**Reviewer Confidence:**

4: Quite sure. I tried to check the important points carefully. It's unlikely, though conceivable, that I missed something that should affect my ratings.

---

> ### Author Rebuttal · Authors · 2023-08-29
>
> ## Response to Reviewer PVJR
>
> Thank you for your comments and questions! We appreciate that the reviewer mentioned the our method effective and leads to the state-of-the-art performances. In this discussion we hope that we can further address all your concerns as well as answer the questions.
>
> ### 1. Response to Reason to Reject 1: Motivations for using diffusion model and Comparison of using other model(T5) of similar parameter size.
>
> As replied to the reviewer B9MB, we would like to put the _“intermediate reasoning step”_ by generative a hypothetical document, mainly motivated by the success of the chain-of-thought (CoT) [1] and its relevant IR applications [2-3].
> The motivation for using diffusion models to generate such a pseudo document Is that we regard this process as “semantic inpainting”, similar to the image inpainting, where other remaining document contents need to be infilled given a query content. In such inpainting task, the diffusion model has shown largely successful results and performances [4-8]. In addition, diffusion models are increasingly applied to text generation process recently, showing improved performances [9-11].
>
> Of course, we could apply the generative models such as LLM such as GPT3,  LLaMA, and PaLM to perform such the semantic inpainting task. But, given the recently emergent interests in diffusion models, it would be quite valuable and interesting to apply the diffusion model and explore by checking whether the currently available model works well on this addressed semantic inpainting task.
> In addition to this general motivation, there are some technical motivations in using diffusion models as follows:
>
> + The diffusion model is parameter-efficient; in the work of Diffusion-LM[9], DiffuSeq[10], SeqDiffuSeq[11], the small-scale diffusion models outperform some GPT variants on different generative tasks.
>
> + Our semantic inpainting task does not necessarily produce grammatically perfect sentences, which making that the use of LLMs is not mandatory:
> While LLMs can also perform generative tasks and even generate sentences that are of higher quality in terms of syntax and grammar, these capabilities may not be mandatory for our addressed semantic inpainting task. In our task, pseudo documents are not necessarily grammatically perfect but required to provide semantically enriched results, which are helpful to the next stage of the retrieval; In fact, our example pseudo documents generated by diffusion models are somehow grammatically flawed as shown in Appendix A.
>
> Of course, we are also quite interested in applying LLMs to the semantic inpainting task. In the future, we will compare the results of using LLMs and diffusion models in generating pseudo-documents, in any extended version of the paper.
>
> On the other hand, given your additional question _"How would DiffusionRet work with a generative language model (like T5) of the same parameter size?"_, we trained a T5-base (220M, twice the parameters of the diffusion model) which are  small language models (SLM), for generating pseudo documents on MS MARCO 100k dataset and show the retrieval results.
>
> As shown in the Table below, the results confirm that the use of SLM does not show good performance and diffusion models are quite parameter-efficient. Unlike SLM, we believe that the use of LLMs will gain much improvements with its strong generation quality, and thus the comparison of LLMs with diffusion models is interesting and valuable as mentioned above.
>
> | Gen-Model | source | Hits@1 | Hits@10 | Params |
> | ---- | ---- | :----: | :----: | :----: |
> | - | query | 40.47 | 71.15 | 0 |
> | T5-base | gen-doc | 19.53 | 45.63 | 220M |
> | T5-base | query+gen-doc | 45.79 | 76.05 | 220M |
> | Diffusion | gen-doc | 80.69 | 93.35 | 103M |
> | Diffusion | query+gen-doc | 85.29 | 96.62 | 103M |
>
> If given a chance, we would like to strengthen our motivation in diffusion models based to the above responses, add detailed comparison results between SLM and diffusion models, and make further statement that the comparison with LLMs is interesting and valuable to invoke the future work.
>
> ### 2. Response to Reason to Reject 2: Discussion of the difference between DiffusionRet's N-gram-based Generative Retrieval and SEAL.
>
> We are sorry that our current presentation does not clearly reveal the difference between n-gram generative retrieval and SEAL [12]. As you mentioned, the length of n-gram is one of differences, but there are other distinguishing parts  from SEAL [12], while some of them are presented in our papers.
>
> + Types of a source sequence is different between SEAL and our method; while SEAL uses short queries, our n-gram generative retrieval deploys _“pseudo documents”_ as for source sequence. The results of SEAL and our method are compared in Table 2, which provide evidence that the use of pseudo documents as source part is important in improving the retrieval performance. documents made a major contribution to the performance improvement. In fact, this training difference is mentioned in **Section 4.4-Training Retrieval Model**.
>
> + SEAL extensively uses the query generation as data augmentation by uniformly sampling spans from the same documents, where the sampled spans are other queries of the same document. In contrast, our N-gram generative retrieval has no such data augmentation based on query generation, as our method only uses a single instance of pseudo document, without more examples per document. This difference is presented in Appendix H:
> > _It is noted that SEAL uses uniformly sampled spans from documents in the collection as “unsupervised” samples to train, aiming to expose the model to more possible pieces of evidence. In our generation setting, our generated documents contain semantically-rich contextual information, we do not need such additional data._
>
> To clarify this difference between SEAL and our N-gram generative retrieval, if given a chance, we would like to add the key difference clearly in the main text, and revise Appendix H such that data augmentation is not used in our method.
>
> To further clarify the comparison with original version of SEAL, we would like to add further experiments by directly evaluating SEAL using short queries as source part, both MS MARCO 100k and NQ 320k, varying the length of n-grams, in any extended version.
>
> ### 3. Response to Reason to Reject 3: Formatting issue
>
> Thank you for pointing out the formatting issue. We use latex to control the format and have checked the tex file to ensure that there are no redundant blank lines.
>
> ### 4. Response to Q1: Discussion about generation quality and inference speed of diffusion model.
>
> Thank you for your insightful question. It would be valuable and interesting to make the detailed comparison between the diffusion model and SLM (i.e., T5) in terms of not only the retrieval performance but also other generation quality and inference speed. If given a chance, we would definitely add more ablation studies comparing between the diffusion model and T5, both in terms of the inference speed and generation quality, as you raised. Here, let us mention the responses considerable on the raised issues.
>
> + Inference speed: Because diffusion models require thousands of diffusion steps to sample the data during the inference process, the number of sampling steps or the sampling algorithm has a decisive influence on the inference speed rather than the model size. As emphasized in **Limitations**, the inference time of diffusion models is relatively slow, and enhancing their inference speed has gained significant research attention. Some specialized works have achieved significant success in this field (e.g., Consistency Model[13], LatentOps[14]), making it a good direction for future work. The comparison in inference time between T5 and diffusion model would be more interesting, when those recently-advanced diffusion models [13-14] are incorporated, which will be worth to be explored in any extended work of this paper.
>
> + Generation quality: While there are various metrics for evaluating the quality of model generated text, we need to concern the specific demand of our semantic inpainting task (i.e., generating pseudo documents). First, we do not need to generate sentences that are of very high quality in syntax or grammar, since our goal is to inject rich semantic information into the input of the retrieval model. Second, in a two-stage model design, the second-stage retrieval model has tolerance for the generation errors of the first stage (TOME[3] also has statements and experimental support for this theory, see the second contribution point in Introduction and Section 5.3.2).
>
> Therefore, our goal in the first stage is to improve the retrieval performance, the evaluation on the generated samples needs to be designed to carefully capture its impact on the second stage. Designing such evaluation metric would be quite important and interesting topic to be explored in the future work.
> In the meanwhile, for reference, we have used he ROUGE score of the generated results of diffusion model in **Appendix F** comparing to the ground-truth one. Here, for convenience, let us copy the table results from Appendix F as follows.
>
>
> |Set|Rouge-1|Rouge-2|Rouge-L|
> |----|:----:|:----:|:----:|
> |train|0.696|0.537|0.677|
> |dev|0.322|0.123|0.257|
>
>
>     [1] Wei, Jason, et al. "Chain-of-thought prompting elicits reasoning in large language models." Advances in Neural Information Processing Systems 35 (2022): 24824-24837.
>     [2] Trivedi, Harsh, et al. "Interleaving retrieval with chain-of-thought reasoning for knowledge-intensive multi-step questions." Proceedings of the 61st Annual Meeting of the Association for Computational Linguistics, 2023.
>     [3] Ren, Ruiyang, et al. "TOME: A Two-stage Approach for Model-based Retrieval.". Proceedings of the 61st Annual Meeting of the Association for Computational Linguistics, 2023.
>     [4] Saharia, Chitwan, et al. "Palette: Image-to-image diffusion models." ACM SIGGRAPH 2022 Conference Proceedings. 2022.
>     [5] Lugmayr, Andreas, et al. "Repaint: Inpainting using denoising diffusion probabilistic models." Proceedings of the IEEE/CVF Conference on Computer Vision and Pattern Recognition. 2022.
>     [6] Dhariwal, Prafulla, and Alexander Nichol. "Diffusion models beat gans on image synthesis." Advances in neural information processing systems 34 (2021): 8780-8794.
>     [7] Li, Wenbo, et al. "Sdm: Spatial diffusion model for large hole image inpainting." arXiv preprint arXiv:2212.02963 (2022).
>     [8] Parida, Sibam, et al. "Survey on Diverse Image Inpainting using Diffusion Models." 2023 2nd International Conference on Paradigm Shifts in Communications Embedded Systems, Machine Learning and Signal Processing (PCEMS). IEEE, 2023.
>     [9] Li, Xiang, et al. "Diffusion-lm improves controllable text generation." Advances in Neural Information Processing Systems, 2022.
>     [10] Gong, Shansan, et al. "Diffuseq: Sequence to sequence text generation with diffusion models." arXiv preprint arXiv:2210.08933 (2022).
>     [11] Yuan, Hongyi, et al. "Seqdiffuseq: Text diffusion with encoder-decoder transformers." arXiv preprint arXiv:2212.10325 (2022).
>     [12] Bevilacqua, Michele, et al. "Autoregressive search engines: Generating substrings as document identifiers." Advances in Neural Information Processing Systems 35 (2022): 31668-31683.
>     [13] Song, Yang, et al. "Consistency Models." arXiv preprint arXiv:2303.01469 (2023).
>     [14] Liu, Guangyi, et al. "Composable text controls in latent space with odes." arXiv preprint arXiv:2208.00638 (2022).

---

### Official Review · Reviewer_B9MB · 2023-08-12

**Soundness:** 3

**Excitement:**

3: Ambivalent: It has merits (e.g., it reports state-of-the-art results, the idea is nice), but there are key weaknesses (e.g., it describes incremental work), and it can significantly benefit from another round of revision. However, I won't object to accepting it if my co-reviewers champion it.

**Paper Topic And Main Contributions:**

In this paper, the authors propose to leverage a diffusion-based two-stage generative retriever. It first leverages a diffusion model to generate pseudo documents based on input queries to address the gap between indexing and retrieving tasks. Then it performs a doc-ngrams retrieval with the generated documents. Experiments on two public datasets verify the effectiveness of the proposed model.

**Reasons To Accept:**

1. The authors integrate the diffusion model into the generative retrieval.
2. The authors conduct extensive experiments to demonstrate its effectiveness.

**Reasons To Reject:**

1. The motivation is not reasonable. As claimed in abstract, i.e., "Generative retrieval suffers from the lack of the intermediate reasoning step, caused by the manner of merely using a query to perform the hierarchical classification. ", I have the following questions:
   - The lack of intermediate reasoning step is not necessarily a weakness, as it implies an end-to-end retrieval, which on the contrary should be an advantage.
   - Additionally, generative retrieval is not a hierarchical classification but a query2docid generation task, where docid can be atomic or ngrams like SEAL and this paper.
   - The authors should give some evidences to support this claim.
2. The model design to address the pretrain-finetune discrepancy lacks justification. Existing studies such as DSI and NCI have adopted query generation to address this discrepancy, while this paper applies document generation instead. The reason, necessity, and advantage of doing so compared to query generation are not provided in the paper.
3. Experiment 5.3 is essentially evaluating the ability of the adopted diffusion model (i.e., SeqDiffuSeq) to handle unseen documents. But for retrieval task, these documents are still visible and used for training. As such, this experiment cannot measure the retrieval performance of the proposed model for unseen documents.

**Reproducibility:**

4: Could mostly reproduce the results, but there may be some variation because of sample variance or minor variations in their interpretation of the protocol or method.

**Reviewer Confidence:**

5: Positive that my evaluation is correct. I read the paper very carefully and I am very familiar with related work.

---

> ### Author Rebuttal · Authors · 2023-08-29
>
> ## Response to Reviewer B9MB
>
> Thanks for taking you time to review this manuscript. We appreciate that the reviewer commenting on the effectiveness of our work. Thank you again for your comments and the issues raised. However, we also believe that there are misunderstandings about some certain aspects of our work, in this discussion we will further explain all your concerns and correct some misconceptions.
>
> ### 1. Response to Reason to Reject 1: Advantages of intermediate reasoning step and Explanation of the usage of term hierarchical classification
>
> Thank you for raising insightful issues.
>
> **For the intermediate reasoning step issue.**
> Explicitly putting _“intermediate reasoning step”_ is mainly motivated by the success of the chain-of-thought (**CoT**) on various NLP tasks and the CoT-based retrieval and QA application [1,2]. Particularly for our addressed retrieval task, the use of intermediate reasoning step is directly motivated by the recently published two-stage generative retrieval [3].
>
> As in [3], the generative retrieval can be equipped well with the intermediate reasoning step, possibly even in an end-to-end manner. Overall, we strongly believe that the generative retrieval itself has much advantages and potential to be extended. But, our criticism made in this statement is mainly targeted the early works of the generative retrieval that maps directly a query to docids, which does not explicitly put the intermediate reasoning step (or intermediate content), not all generative retrieval methods, particularly not the up-coming generative retrieval methods that will be extended and improved. Here, we want to mention that when the generative retrieval does not use the intermediate reasoning step, its performance may be limited, given the existing successful results using CoT and other similar works, including [3].
>
> Please understand our result provides some evidence that the intermediate reasoning step is helpful in improving the performance, with the stronger improvements than [3]. We believe that our result is positively affects the extension of generative retrieval. While the intermediate reasoning step is currently based on the diffusion model in our current paper, we are also really interested in extending the generative retrieval from the currently observed evidence.
>
> But, we agree with your concern that we should be more careful in making this the criticism, not attacking all generative retrievals. To clarify that the limitation only belongs to the specific type of generative retrieval (but currently widely used), we would revise the statement and paragraph in a very careful way without making miss-understanding our intention, in any extended version. Also, in any further chance, we would like to add more references for motivating our work, including [2] which was missed in the current version of the paper.
>
> **For the use of the term hierarchical classification issue.**
> As in the response above, our statement here concerns mainly a specific type of the generative retrieval, which is the early work of DSI (but still widely used as the basic framework to our knowledge), not all possible generative retrieval methods. Under such a specific type of generative retrieval, we view it as the hierarchical classification, largely based on [4], which interprets the hierarchical document classification as a sequence-to-sequence task; In [5], a hierarchical clustering process is deployed to assign semantically structured identifiers.
>
> In fact, our intention here is not to reveal the _“hierarchical classification”_ itself but to strongly emphasize that the intermediate reasoning step is not involved in (some) generative retrieval methods, in a way of that it directly goes into the classification process, not “generating” the intermediate contents.
> Again, we also aware your concern that the statement is also needs to be carefully provided. To clarify our intended meaning, If given the chance, we would like to rewrite the statement in a much proper way (such as removing the problematic term _“… mapping a query directly to docids or index terms, …”_, or clearly mention that only specific type of generative retrieval can be seen as the hierarchical classification with the reference [4]).
>
> ### 2. Response to Reason to Reject 2: Evidence for addressing pretrain-finetune discrepancy.
>
> First of all, we are really sorry that DSI-QG [6], the reference in the paragraph when mentioning the term “pretraining-finetuning discrepancy” in Section 1, is wrongly cited; its correct one is TOME [3]. In other words, we have the citation-incorrect sentence in the 2nd paragraph in Section 1:
>
> > _As noted in (Zhuang et al., 2022), however these docid-like symbols are only seen during the finetuning but unseen at the pretraining stage, thus causing the pretrain-finetune discrepancy, which ..._
>
> which should be revised to the sentence below:
>
> > _As noted in (Ren et al., 2023), however these docid-like symbols are only seen during the finetuning but unseen at the pretraining stage, thus causing the pretrain-finetune discrepancy, which …_
>
> We far believe that the wrong citation makes the confusion in understanding the intended meaning of “pretraining-finetuning discrepancy”. We are deeply sorry about this error again.
>
> In fact, the issue of the pretraining-finetuning discrepancy of the generative retrieval is NOT newly addressed in this paper, instead being raised in [3]. To clarity this, let us allow to copy the corresponding snippet of paragraph of TOME [3] which mentions the pretraining-finetuning discrepancy:
>
> >_…
> Such docids are not adequately captured in the pretraining stage of the generative PLM, thus limiting PLM’s capabilities for generative prediction (e.g., unseen docids during pre-training). This creates a discrepancy between the pre-training and fine-tuning phases
> …_
>
> Thus, our statement of the _“pretraining-finetuning discrepancy”_ is a re-expression of the above cited part [3], thus having the exactly same meaning as that of [3].
>
> On the other hand, the terminology “pretraining-finetuning discrepancy” has also been introduced in XLNet [7], when criticizing BERTs, meaning that [MASK] token is ONLY seen during the pretraining but is NOT seen in downstream tasks during finetuning. This pretraining-finetuning discrepancy of BERT is quite similar to that of the generative retrieval, as two cases are all caused by introducing artificially-designed tokens (i.e., [MASK] and docids).
>
> Because generative retrieval relies on the use of _“pretrained”_ transformers (i.e., encoder-decoder models), we expect that the effect of using the transfer learning from the pretrained model to the finetuned one in the downstream retrieval task will be substantially enlarged and improved when the pretraining-finetuning discrepancy (currently caused by docids) is somehow reduced or relaxed.
>
> This motivates us to make the second stage of the retrieval based on the N-gram generative retrieval, without using docid-like symbols, although we adopt the existing SEAL model [8].
>
> While we don’t clearly provide the comparison results with details, the potential advantage of reducing the pretraining-finetuning discrepancy from using n-grams is the training efficiency of the retrieval model; in our preliminary experiments, the n-gram generative retrieval is much quickly trained only using single training example per each document in the collection, comparing to DSI-QG [6]. Given our limited time, we couldn’t prepare the comparison results on that. If given change, we would definitely provide the results of the comparing training costs between our N-gram retrieval model and DSI-QG.
>
> We also want to clarify that the discrepancy addressed by DSI-QG [6] and NCI [9] using the query generation is NOT the aforementioned pretraining-finetuning discrepancy, but the data distribution mismatch [6]; the long text of a document is trained during the indexing time, but a short query text is provided during the inference time. In fact, this mismatch issue is somehow handled by putting “intermediate reasoning step” in our DiffusionRet, as we generate a long pseudo document, whose length is similar to that observed during indexing time.
> Comparison with those approaches using query generation is also interesting issue. As reported in Table 1-2, our DiffusionRet shows better performances than DSI-QG [6] and NCI [9] . Of course, the experiment results only provide positive evidence that the putting _“intermediate reasoning step”_ is helpful in improving the performance, not logically determining a better one between the query generation and the intermediate reasoning step.
> We believe that the generative retrieval equipped with query generation is also advantageous, as it provides the end-to-end retrieval mechanism. Integrating these two directions for the query generation and the intermediate reasoning step would be interesting to be explored.
>
> To clearly resolve your concern, if given a chance, we would like to revise our paper in any extended version, as follows:
>
> 1. Correcting the wrongly-cited reference to TOME [3]
> 2. Clarifying that the pretraining-finetuning discrepancy is introduced in [3] (as reference, the terminology is used in [7] similarly)
> 3. Clarifying that the pretraining-finetuning discrepancy is different from the distribution mismatch issue addressed by the generative retrievals trained with query generation like DSI-QG [6] and NCI [9]
> 4. Stating that generative retrievals trained with the query generation are on the valuable direction, so being complementary with the direction of putting the intermediate reasoning step.
> 5. Presenting the comparison of training costs between the N-gram generative retrievals and DSI-QS [6] (or NCI [9]), to show that reducing pretraining-finetuning discrepancy is helpful to improve the training efficiency.
>
> ### 3. Response to Reason to Reject 3: Explanations for Experiment 5.3.
>
> We are sorry for making this confusion. Here, we want to mainly check the robustness the diffusion model to other unseen 200k queries, rather than the retrieval model itself. In other words, the diffusion model is trained only using 100k examples, but being expanded to other 200k unseen queries, thereby making the enlarged search space, (i.e., the search space is the sets of 100k and 300k documents in Table 1 and Table 4, respectively.) That is why the section title is “expansion of diffusion model to unseen documents on MS MARCO 300k”, thereby focusing on the diffusion model.
>
> But, as you worried, the retrieval models are not fully unseen; our retrieval model has the subtle difference from the MS MARCO 100k setting; While the retrieval model is trained using 300k, 200k pseudo documents may be different from 100k ones in quality, because the diffusion model is not trained on those 200k queries.
>
> Generally, we believe that your raised issue and experiment setup is also important and valuable. In given a chance, we would like to provide the retrieval performance of the fully unseen setting in MS MARCO 300k in any extended version, where n-gram generative retrieval only trained over 100k documents is extensively applied over 300k documents.
>
> We deeply appreciate your efforts in reviewing our paper. Hope this discussion could address all your concerns. Wish you all the best.
>
>     [1] Wei, Jason, et al. "Chain-of-thought prompting elicits reasoning in large language models." Advances in Neural Information Processing Systems 35 (2022): 24824-24837.
>     [2] Trivedi, Harsh, et al. "Interleaving retrieval with chain-of-thought reasoning for knowledge-intensive multi-step questions." Proceedings of the 61st Annual Meeting of the Association for Computational Linguistics, 2023.
>     [3] Ren, Ruiyang, et al. "TOME: A Two-stage Approach for Model-based Retrieval.". Proceedings of the 61st Annual Meeting of the Association for Computational Linguistics, 2023.
>     [4] Risch, Julian, Samuele Garda, and Ralf Krestel. "Hierarchical document classification as a sequence generation task." Proceedings of the ACM/IEEE Joint Conference on Digital Libraries, 2020.
>     [5] Tay, Yi, et al. "Transformer memory as a differentiable search index." Advances in Neural Information Processing Systems 35 (2022): 21831-21843.
>     [6] Zhuang, Shengyao, et al. "Bridging the gap between indexing and retrieval for differentiable search index with query generation." arXiv preprint arXiv:2206.10128 (2022).
>     [7] Yang, Zhilin, et al. "Xlnet: Generalized autoregressive pretraining for language understanding." Advances in neural information processing systems 32 (2019).
>     [8] Bevilacqua, Michele, et al. "Autoregressive search engines: Generating substrings as document identifiers." Advances in Neural Information Processing Systems 35 (2022): 31668-31683.
>     [9] Wang, Yujing, et al. "A neural corpus indexer for document retrieval." Advances in Neural Information Processing Systems 35 (2022): 25600-25614.

---

### Meta-Review · Area_Chair_Uoj3 · 2023-09-23

**Recommendation:** 4

**Metareview:**

All the reviewers have agreed that it is interesting to see the integration of diffusion models into generative retrieval and the improvement gained.

However, common concerns have also been raised that the motivation is somehow weak and some discussion on the relations and differences between the proposed method and its competitors are missing.

During the rebuttal period, the authors have only partly addressed these concerns, leaving room for further improvement.

---

### Decision · Program_Chairs · 2023-10-07

**Decision:**

Accept-Findings

**Comment:**

All the reviewers have agreed that it is interesting to see the integration of diffusion models into generative retrieval and the improvement gained.

However, common concerns have also been raised that the motivation is somehow weak and some discussion on the relations and differences between the proposed method and its competitors are missing.

During the rebuttal period, the authors have only partly addressed these concerns, leaving room for further improvement.